# Microbial Fermentation and Its Role in Quality Improvement of Fermented Foods

**Ranjana Sharma** [1,2,†], **Prakrati Garg** [1,2,†], **Pradeep Kumar** [1,2], **Shashi Kant Bhatia** [3]  **and Saurabh Kulshrestha** [1,2,*]

1   Faculty of Applied Sciences and Biotechnology, Shoolini University of Biotechnology and Management Sciences, Bajhol, Solan 173229, Himachal Pradesh, India; ranjanasharma@shooliniuniversity.com (R.S.); prakrati@shooliniuniversity.com (P.G.); pradeep.kumar@shooliniuniversity.com (P.K.)

2   Center for Omics and Biodiversity Research, Shoolini University of Biotechnology and Management Sciences, Bajhol, Solan 173229, Himachal Pradesh, India

3   Biotransformation and Biomaterials Lab, Department of Microbial Engineering, College of Engineering, Konkuk University, Hwayang-dong, Gwangjin-gu, Seoul-05029, Korea; shashibiotechhpu@gmail.com

*   Correspondence: saurabh_kul2000@yahoo.co.in or sourabhkulshreshtha@shooliniuniversity.com; Tel.: +91-962-503-3405

†   Contributed equally.

**Abstract:** Fermentation processes in foods often lead to changes in nutritional and biochemical quality relative to the starting ingredients. Fermented foods comprise very complex ecosystems consisting of enzymes from raw ingredients that interact with the fermenting microorganisms' metabolic activities. Fermenting microorganisms provide a unique approach towards food stability via physical and biochemical changes in fermented foods. These fermented foods can benefit consumers compared to simple foods in terms of antioxidants, production of peptides, organoleptic and probiotic properties, and antimicrobial activity. It also helps in the levels of anti-nutrients and toxins level. The quality and quantity of microbial communities in fermented foods vary based on the manufacturing process and storage conditions/durability. This review contributes to current research on biochemical changes during the fermentation of foods. The focus will be on the changes in the biochemical compounds that determine the characteristics of final fermented food products from original food resources.

**Keywords:** food fermentation; enzymes; fermenting microorganisms; biochemical changes

## 1. Introduction

Fermentation is a process that helps break down large organic molecules via the action of microorganisms into simpler ones. For example, yeast enzymes convert sugars and starches into alcohol, while proteins are converted to peptides/amino acids. The microbial or enzymatic actions on food ingredients tend to ferment food, leading to desirable biochemical changes responsible for the significant modification to the food. Fermentation is a natural way of improving vitamins, essential amino acids, anti-nutrients, proteins, food appearance, flavors and enhanced aroma. Fermentation also helps in the reduction of the energy needed for cooking as well as making a safer product [1,2]. Therefore, microorganisms' activity plays a significant role in the fermentation of foods by showing changes in the foods' chemical and physical properties. Fermented foods have several advantages [3,4]:

(1)   Fermented foods have a longer shelf life than the original foods.

(2)   The enhancement of organoleptic properties; for example, cheese has more enhanced organoleptic properties in terms of taste than its raw substrate viz. milk.

(3)　The removal of harmful/unwanted ingredients from raw materials—for example, during garri preparation, there is a reduction in the poisonous cyanide content of cassava, and the flatulence factors in soybeans are removed by fermentation.

(4)　The enhancement of nutritional properties due to the presence of fermenting microorganisms. For example, yeast in bread and yeast and lactic acid bacteria in garri add to its nutritive quality.

(5)　The fermentation process reduces the cooking time of food. For example, West African food, i.e., Ogi (prepared from fermented maize), and soybean products.

(6)　The fermented products consist of higher in vitro antioxidant capacity. For example, fermented milk and yogurt consist of higher antioxidant properties compared to milk, as there is a release of biopeptides that follow the proteolysis of milk proteins, particularly α-casein, α-lactalbumin, and β-lactoglobulin.

The composition of the substrates used and the fermenting microorganisms are the major factors that influence fermented food. Moreover, food treatment and the length of fermentation during processing also affect food fermentation [5]. For all the fermented foods and beverages that have been identified, lactic acid bacteria (L.A.B.) is the dominant microbiota, which has been considered the most critical part contributing to beneficial effects in fermented foods/beverages [6]. The fermenting microorganisms mainly involve L.A.B. like *Enterococcus*, *Streptococcus*, *Leuconostoc*, *Lactobacillus*, and *Pediococcus* [6] and yeasts and molds viz. *Debaryomyces*, *Kluyveromyces*, *Saccharomyces*, *Geotrichium*, *Mucor*, *Penicillium*, and *Rhizopus* species [7–10]. The fermentative sugar pathway for *Lactobacillus* and yeasts is mentioned in Figure 1. A list of some of the most commonly prepared fermented foods/beverages with their fermenting microorganisms is also discussed in Table 1. Despite adding beneficial effects during fermentation, microorganisms in food also help prevent many harmful chemicals and microorganisms during the fermentation process. These microorganisms are also responsible for the production of new enzymes that assist with digestion.

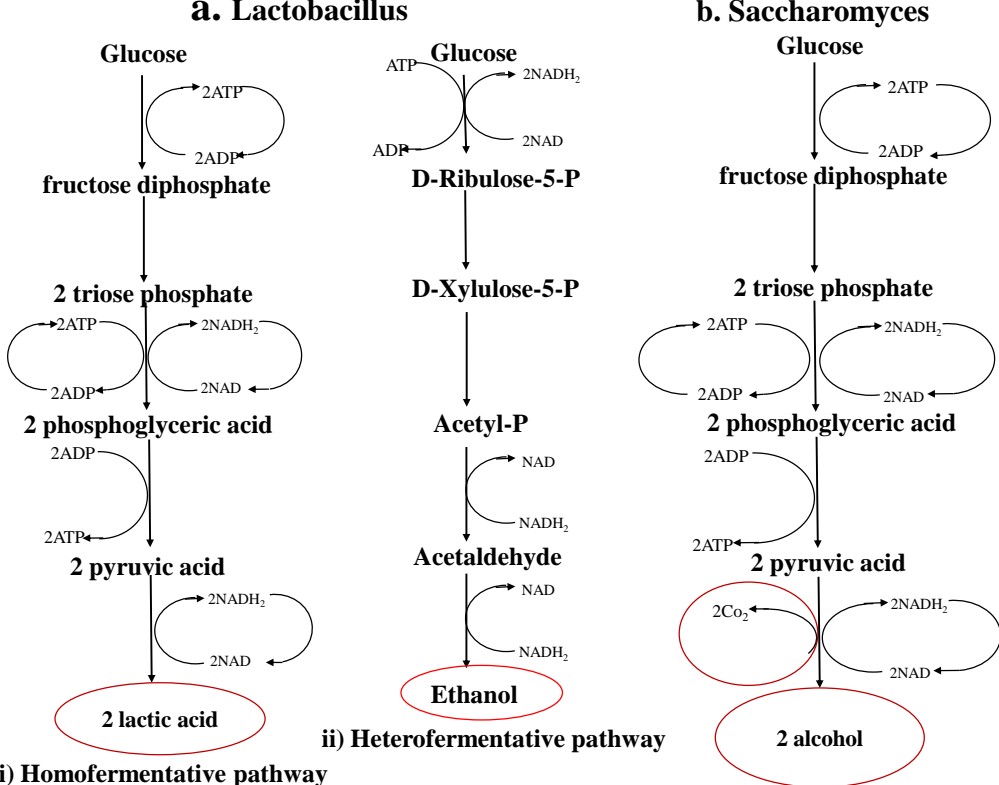

**Figure 1.** Sugar metabolism by *Lactobacillus* and *Saccharomyces* as representatives of L.A.B. and yeasts [11,12].

**Table 1.** Some of the most commonly prepared fermented foods/beverages with their fermenting microorganisms.

| Fermented Foods/Beverages | Substrates Used | Microorganisms Involved in Fermentation |
|---|---|---|
| Dairy products<br><br>Curd, Yogurt, Cheese, Yakult, Kefir | Milk and milk casein | *Lactobacillus bulgaricus, Lactococcus lactis, L. acidophilus, L. cremoris, L. casei, L. paracasei, L. thermophilus, L. kefiri, L. caucasicus, Penicillium camemberti, P. roqueforti, Acetobacter lovaniensis, Kluyveromyces lactis, Saccharomyces cerevisiae* |
| Vegetable products<br><br>Kimchi, Tempeh, Natto, Miso, Sauerkraut | Soybean, cabbage, ginger, cucumber, broccoli, radish | *Leuconostoc mesenteroides, Aspergillus* sp., *Rhizopus oligosporus, R. oryzae, L. sakei, L. plantarum, Thermotoga* sp., *L. hokkaidonensis, L. rhamnosus, Rhodotorula rubra, Leuconostoc carnosum, Bifidobacterium dentium, Enterococcus faecalis, Weissella confusa, Candida sake* |
| Cereals<br><br>Bahtura, Ambali, Chilra, Dosa, Kunu-Zaki, Marchu | Wheat, maize, sorghum, millet, rice | *L. pantheris, L. plantarum, Penicillium* sp., *S. cerevisiae, L. mesenteroides, E. faecalis, Trichosporon pullulans, Pediococcus acidilactici, P. cerevisiae, Delbrueckii hansenii, Deb. tamari* |
| Beverages<br><br>Wine, Beer, Kombucha, Sake | Grapes, rice, cereals | *Aspergillus oryzae, Zygosaccharomyces bailii, S. cerevisiae, Acetobacter pasteurianus, Gluconacetobacter, Acetobacter xylinus, Komagataeibacter xylinus* |
| Meat Products<br><br>Sucuk, Salami, Arjia, Jama, Nham | Meat | *L. sakei, L. curvatus, L. plantarum, Leuconostoc carnosum, Leuconostoc gelidium, B. licheniformis, E. faecalis, E. hirae, E. durans, Bacillus subtilis, L. divergens, L. carnis, E. cecorum, B. lentus* |

## 2. Enhancement of Nutritional Quality in Fermented Foods by Microorganisms

It has been known that fermented foods are more nutritious than their unfermented counterparts [13]. The increased nutritional value in fermented foods is due to the fermenting microorganisms present in them, and the three different ways of fermentation by microorganisms are as follows:

Microorganisms are both catabolic and anabolic, break down complex compounds, and synthesize complex vitamins and other growth factors [14].

Indigestible substances liberate the nutrients that are locked into plant structures as well as cells. This event occurs especially with individual seeds and grains. In the milling process, cellulosic and hemicellulosic structures surrounding the endosperm (viz., rich in proteins and digestible carbohydrates) have been physically ruptured to release nutrients. Crude milling is used in less developed regions to extract nutritional contents, but it is inadequate to release full nutritional value from the plant products. Even after the cooking process, a few of the bounded nutrients remain inaccessible to the human digestive system. At the same time, this issue can be resolved by certain bacteria, molds, and yeasts that decompose or breaks the cell walls and indigestible coatings of these products both physically and chemically [13].

A different mechanism to increase plant material's nutritional properties is through enzymatic degradation of polymers that are not digested by humans into simple sugars and their derivatives like cellulose, hemicelluloses, and a similar form of polymers. Using microbial enzymes, the cellulose-containing substrates in fermented foods can be improved for human consumption [15]. Many cereal foods are low in their nutritional content and are consumed as an essential staple diet for poor people. However, L.A.B. and yeast fermentation were observed to enhance nutritional content and food digestibility. The fermentation process also increases the microbial enzyme activity as it provides an acidic environment at temperature 22–25 °C [16]. The critical function of enzymatic hydrolysis in fermented foods includes a reduction in levels of anti-nutrients viz. tannins and phytic acid (degradation with the help of phytases), resulting in enhanced bioavailability of simple sugars or polysaccharides (amylases), proteins (proteases), free fatty acids (lipases), and iron.

## 3. Effects of Lactic Acid Fermentation on the Nutritional Aspects of Food

The main factors contributing to food's nutritional value include its digestibility and the number of vital nutrients present. Both nutrients, as well as digestibility, may be improved by the process of fermentation. During the fermentation process, the fermented microorganisms' enzymes may initially digest the macronutrients [2]. The several ways by which the nutritional quality of food can be affected by fermentation include increasing the amount and bioavailability of nutrients and enhancing nutrient density. The latter may be achieved by synthesizing promoters for absorption, the degradation of anti-nutritional factors, influencing the uptake of nutrients by the mucosa, and pre-digestion of individual food components [1]. The solubility of proteins and the availability of some micronutrients and limiting amino acids are enhanced by the process of lactic acid fermentation [17]. By this process, tannins (50%), phytates, and oligosaccharides (90%) are also reduced [18]. There can be a direct or indirect nutritional impact of fermented foods on nutritional diseases. The fermentation process of food has a direct curative effect [19]. Likewise, food fermentation contributes directly to consumers' health by increasing the number of available vitamins such as niacin, thiamine, folic acid, or riboflavin [3]. It also enhances iron utilization through the breakdown of complex substances into inorganic iron with vitamin C [1].

Food fermentation increases mineral and trace elements' bioavailability by reducing non-digestible material in plants such as glucuronic and polygalacturonic acids, cellulose, and hemicelluloses [20]. It also reduces serum cholesterol by inhibiting cholesterol synthesis in the liver and dietary and endogenous cholesterol absorption in the intestine [21]. It is robust, stable, and safe for the product, thereby preempting diseases/infections such as diarrhea and salmonellosis [22].

## 4. Enrichment and Changes of Biological Components in Fermented Foods

### 4.1. Vitamins Bio-Enrichment

As a public health measure, nutrients, mainly vitamins, are fortified in some selected, manufactured foods; for example, vitamin D is added to milk and riboflavin during bread production, whereas ascorbic acid (vitamin C) can be fortified in fruit juices (Figure 2). However, this fortification or enrichment process can only be used in the Western world because of its high-cost value. Hence, most countries should use this type of food fermentation for the biological enrichment of foods [23]. There is a deficiency of thiamine (Vitamin B1) caused by using highly polished white rice. This type of rice can cause beriberi, a disease that leads to strokes and paralysis [24]. Infants fed by the thiamine-deficient (lead to beriberi) mothers can also suffer sudden death at three months because of heart failure [25]. Thiamine is synthesized by the microorganisms involved in the tape Ketan fermentation. These microorganisms are also responsible for the restoration of the thiamine level found in unpolished rice [26]. Therefore, this can be of great help to rice-eating individuals.

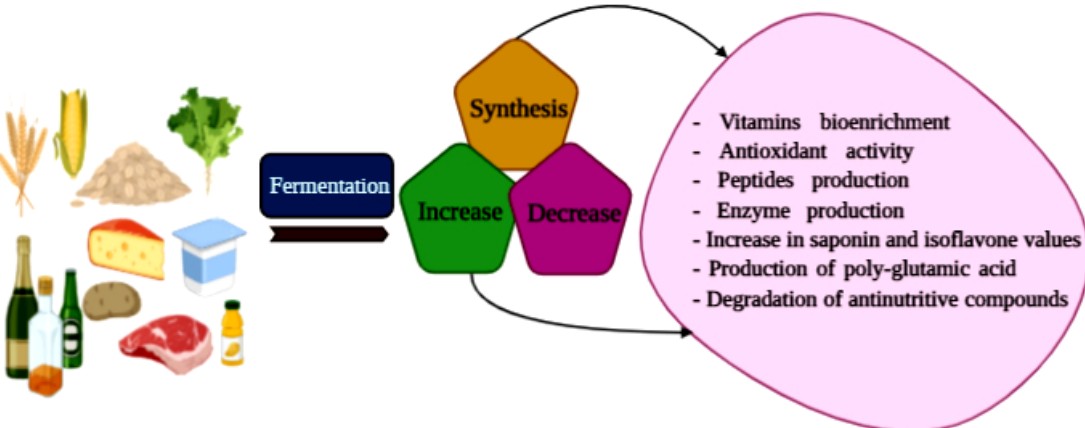

**Figure 2.** Nutritional enhancement in fermented foods.

An Indonesian origin-rich dish known as "Tempe" is prepared by soaking, dehulling, and partial cooking soya beans with the help of *Rhizopus oligosporus* or similar molds [27]. This mold forms a firm cake by knitting cotyledons into slices and being cooked. There is the partial hydrolysis of proteins during the fermentation process; the lipids are hydrolyzed to their constituent stachyose (a tetrasaccharide indigestible by humans), fatty acids, riboflavin doubles, niacin increases by seven-fold, and vitamin B-12, usually absent in vegetarian foods, is synthesized by a fermenting bacterium growing with the essential mold [27]. The manufacturing process of tempe reduces the cooking time and improves the digestibility and texture of many bowls of cereal/legume mixtures [27]. The bacterium, i.e., *Klebsiella pneumoniae* (nonpathogenic strain), is responsible for producing vitamin B-12 when inoculated into Indian idli fermentation [28].

Fermentation of the cactus plant (Agave) juices leads to Mexican pulque production and the oldest alcohol-containing beverage on America's continent [29]. Pulque is very commonly consumed among low-income children of Mexico because of its richness in niacin, thiamine, pantothenic acid, riboflavin, and p-aminobenzoic acid biotin and pyridoxine [30]. An alcoholic beverage, e.g., Kaffir beer, has a thin gruel consistency and pleasant sour taste [31]. Kaffir is a beverage traditionally prepared by the people of Bantu of South Africa with 1 to 8% alcohol content. Kaffir beverage was prepared from malted and unmalted kaffircorn (*Sorghum caffrorum*). The substitution for kaffircorn can be millet or maize. This alcoholic beverage increases riboflavin, and niacin/nicotinic acid nearly doubles, keeping the thiamin level constant during fermentation in people consuming maize [32].

Palm sap is a sweet, plump, milky white suspension of bacteria and yeasts that is a colorless, transparent liquid containing approximately 10 to 12% fermentable sugar [33]. It is consumed in the tropics. This type of wine consists of approximately 83 mg of ascorbic acid/L [34]. In fermented palm wine, thiamine is increased from 25–150 µg/L, pyridoxine: 4–18 µg/L, and riboflavin: 35–50 µg/L. Surprisingly, there is a considerable amount of vitamin B-12 (190 to 280 µg/mL) in palm wine. Palm toddies are the cheapest vitamin B source and play a crucial role in nutrition economically drained in the tropics [35].

### 4.2. Antioxidant Activity

Antioxidant activities in the fermented foods comprise the reducing power assay, 2'-azino-bis (3-ethylbenzo-thiazoline-6-sulfonic acid; A.B.T.S.) and 1,1-diphenyl-2-picryl hydrazyl (D.P.P.H.) radical scavenging activity and total phenol content (TPC) estimation [36,37]. Many Asian soybean fermented foods have antioxidant properties, for example, tungrymbai and bekang (Indian fermented soybean foods) [38], kinema (Indian and Nepal fermented soybean food) [39,40], tempe (Indonesian mold-fermented soybean food) [41], jang and chungkokjang (Korean fermented soybean foods) [42,43], thuanao (Thailand fermented soybean food) [44], natto (Japan fermented soybean food) [45], and douchi (China fermented soybean food) [46]. The antioxidant activity has also been found in yogurt [47] and kimchi [48]. Abubakar et al. determined the antioxidant activity of lactic acid bacteria during fermentation of skim milk via 1,1-diphenyl-2-picrylhydrazyl (D.P.P.H.) assay and observed free radical scavenging activity ranging from 14.7 to 50.8% (*v/v*) after fermentation up to 24 to 72 h, respectively [37]. It has been noticed that the different species of L.A.B. used and fermentation time affected the antioxidant activity significantly ($p \leq 0.05$). The improvement in the quality of fermented sausage in terms of antioxidant value by *L. curvatus* (SR6) and *L. paracasei* (SR10-1) was assessed by [49]. The sausage's antioxidant activity was significantly enhanced by the bacteria, which leads to the overall improvement in the sensory attributes and sausage safety. The D.P.P.H. scavenging activity and reducing power was better in the case of *L. curvatus* SR6 viz. 59.67% ± 6.68% and 47.31% ± 4.62%, whereas O.H. scavenging activity and anti-lipid peroxidation capacity were better with *L. paracasei* SR10-1, i.e., 285.67% ± 2.00% and 63.89% ± 0.93%, respectively.

### 4.3. Peptides Production

The bioactive peptides (B.A.P.s) are produced by proteolytic microorganisms (mostly *Bacillus*) during food fermentation [50,51]. Peptides have some antihypertensive properties [52] and also play a vital functional role as antithrombic [53] and immunomodulatory agents [54]. Due to the lack of large-scale production facilities and soaring costs of the enzymes for protein hydrolysis, the production of B.A.P.s is still below the benchmark. However, microbial fermentation is a more economical and cost-effective method for the production of B.A.P.s. The health properties associated with dairy products are existent in B.A.P.s. To date, four different types of cell envelope protienase (C.E.P.s) have been discovered, which are PrtB from *L. delbrueckii* subsp. *bulgaricus* [55], PrtP from *L. casei* and *L. paracasei* [56], PrtR from *L. rhamnosus*, and *L. plantarum* [57], and PrtH from *L. helveticus* [58]. Generally, most of the *Lactobacillus* species have only one C.E.P., whereas the presence of four different paralogs of PrtH (i.e., PrtH1, PrtH2, PrtH3, and PrtH4) has been found out in *L. helveticus*, and their distribution is strain-dependent [59]. Due to several C.E.P. paralogs and different specificities, *L. helveticus* is the most proteolytic species among the *Lactobacillus* genus, which is solely responsible for the generation of various varieties of B.A.P.s [60].

The activity related to inhibition by angiotensin-converting enzyme (A.C.E.) has also been observed in some milk products previously fermented such as koumiss [61], kefir [62], yogurt [63], cheese [64], fermented milk of camel [65], and fermented products of fish [66]. Bioactive peptides are produced by fermentation of soybean products, which helps prevent and treat various metabolic diseases [67].

### 4.4. Enzymes Production through Microorganisms

Enzymes such as amylase, proteinase, mannase, catalase, cellulose, etc. are generally produced from fermenting microorganisms, especially Bacillus, in Asian soya bean product fermentation of foods that hydrolyze complex substances into simple biomolecules (Table 2) [38]. Carbohydrates producing enzymes viz. like amyloglucosidase, α-amylase, maltase, pectinase, invertase, cellulase, alkaline proteases, lipase, and β-galactosidase are produced from mycelia fungi such as *Amylomyces*, *Actinomucor*, *Aspergillus*, *Mucor*, *Monascus*, *Rhizopus* and *Neurospora* in fermented foods/beverages [68]. The enzyme produced by *A. oryzae* in *koji*, i.e., Taka-amylase A (T.A.A.), has numerous uses in industries [69]. In the Himalayan region, stable, dry, and cake-like amylolytic starter cultures are used to produce alcohol. These starter cultures have mixed yeast strains such as *Saccharomycopsis capsularis*, *S. fibuligera*, and *Pichia burtonii*, increasing amylase [70,71].

**Table 2.** Some essential commercial enzymes used in fermented foods/beverages.

| Substrates | Enzymes | Microbial Source | Enzymatic Action/Process |
|---|---|---|---|
| Dairy | Protease<br>Catalase<br>Lactase | *A. niger*, *A. orzyae* and *B. subtilis*<br>*S. boydii* and *Bacillus* sp.<br>*B. subtilis* | Cheese production<br>Removing $H_2O_2$<br>Lactose-free milk |
| Cereals | Amylase<br>Protease<br>Pentosanase<br>Glucose oxidase<br>Phytase<br>Pullulanase<br>Xylanase<br>Lipases<br>B-glucanase<br>A-acetolactate-decarboxylase<br>Amyloglucosidase<br>Cellulase<br>Pectinase | *B. licheniformis* and *B. subtilis*<br>*A. niger*<br>*Trichoderma* sp.<br>*P. notatum*<br>*A. niger*<br>*B. acidopullulyticus*<br>*A. oryzae* and *B. subtilis*<br>*Aspergillus niger*<br>*B. subtilis*, *A. niger* and *P. funiculosum*<br>*B. subtilis*<br>*A. niger* and *A. flavus*<br>*T. longibrachiatum*<br>*A. niger* | Malting, mashing, liquefaction, and production of flavor esters |

| Substrates | Enzymes | Microbial Source | Enzymatic Action/Process |
|------------|---------|------------------|--------------------------|
| Beverages | Glucose oxidase | *P. notatum* | Clarification of juices |
| | Tannase | *A. niger* | Removing $O_2$ Hydrolysis of esters |
| Meat | Papain | *S. aureus* | Tenderization of meat |
| | Protease | *T. longibrachiatum, A. niger, A. oryzae* and *B. subtilis* | |

The enzyme nattokinase produced by *B. subtilis* present in natto has been observed for its fibrinolytic activity [72,73]. Other bacterial strains isolated from fermented foods like *B. amyloliquefaciens, Vagococcus carniphilus, V. lutrae, P. acidilactici, Enterococcus faecalis, E. faecium,* and *E. gallinarum* also shows fibrinolytic activity [51,74,75]. The SK1-3-7 strain of *Virgibacillus halodenitrificans* isolated from fermented fish sauce [76] also showed fibrinolytic activity.

### 4.5. Increase in Saponin and Isoflavone Values and Poly-Glutamic Acid Production

Isoflavones such as genistein, glycitein, and daidzein are derived from four chemical forms present in soybeans: β-glucoside, aglycones, malonylglucoside, and acetylglucoside [77]. Some Asian fermented soybean products, such as natto and miso, Japan [78], douche and sufu, China [46,79], doenjang and chungkokjang, Korea [80], thuanao, Thailand [44] and tempe, Indonesia [81] hydrolyze isoflavone glucosides into their corresponding aglycones during fermentation. The contents of aglycone and isoflavone, especially Factor-II, were found to increase during fermentation of tempe [82]. Isoflavones in doenjang were also observed to increase the LDL-C receptor activation, capable of preventing vascular diseases [83].

Soybean saponins that are oleanane triterpenoid glycosides in nature are further divided into two types viz., D.D.M.P. (2, 3-dihydro-2, 5-dihydroxy-6-metyl-4*H*-pyran-4-one) and Group A [84]. Beneficial health properties such as suppression of the proliferation of colon cancer cells [85], inhibition of peroxidation activity of lipids [86], and prevention from hypercholesterolemia [87] are some beneficial health effects of D.D.M.P. and their derivatives, Groups B and E saponins. There is an increase in natto's saponin content (generated by *B. subtilis*) [88]. It has been found that there is a high content of *Kinema* in Group B saponins, indicating its goodness to consumers [89].

Poly glutamic acid (P.G.A.) is produced by strains of *Bacillus* sp., such as *B. licheniformis* and *B. subtilis* Asian fermented soybean products [38,90–92], and is not synthesized by proteins present in ribosomes [93]. Due to their biodegradable nature, water-soluble properties, and non-toxicity towards humans, viscous materials of Asian fermented soybean products are safe to consume [94].

### 4.6. Anti-Nutritive Compounds Degradation

Anti-nutritive substances are found in most of the food substrates. They are toxic to human beings and are responsible for limiting the availability of nutrients to the body by decreasing food nutritive value. Anti-nutritive substances may be degraded by microorganisms inhabited in fermented foods, making inconsumable products safe for consumption [95]. Other steps employed during processing, such as grating, dewatering, washing, peeling, fermentation, and roasting, help minimize the cyanide content in the finished products in various African fermented cassava products, gari, and fufu [96]. In some varieties of cassava tubers that are bitter, the presence of cyanogenic glycoside lotaustralin and linamarinis detoxified during traditional methods of fufu and gari production by *Lactobacillus, Streptococcus,* and *Leuconostoc* to produce hydrocyanic acid (H.C.N.) with a low boiling point that escapes during toasting from the dewatered pulp makes the final product nontoxic and safe for consumption [96–98]. However, *R. oligosporus* used in Tempe removes the flatulence from indigestible oligosaccharides such as verbascose and stachyose and converts it into absorbable monosaccharides

and disaccharides [99]. Whereas anti-nutritive compounds are degraded in kinema by *B. subtilis*, the phytic acid reduction is made during idli and rabadi fermentation [95].

*4.7. Biochemical Changes during Cereal Fermentation*

All over the world, cereals are the most crucial carbohydrates, dietary proteins, minerals, fibers, and vitamins [100]. However, the major problem that exists with cereals is their acceptance among consumers in terms of its nutritional quality and sensorial properties of their products compared to milk and milk products. This is due to the low content of proteins, lack of some essential amino acids (for example, lysine), low availability of starch, presence of anti-nutrients (such as polyphenols tannins, and phytic acid), and the coarse nature of the cereals [101].

Several methods have been employed to alleviate the nutritional contents of cereals. For example, improvement through genetic modifications and supplementation of amino acids with concentrates of proteins or alternate rich sources of proteins such as defatted/legumes oilseed meals of cereal grains. In addition to this, various processing technologies such as sprouting, cooking, fermentation, and milling enhance the nutritive quality of cereals, among which fermentation is the best one [101]. Natural fermentation of cereals generally decreases the level of carbohydrates with non-digestible oligo and polysaccharides, improving vitamin B group availability and the synthesis of amino acids [102]. Natural fermentation promotes enzymatic degradation of phytates by providing optimum pH conditions present in a complex form with polyvalent cations such as zinc, iron, magnesium, and calcium [1]. This decrease in phytate can increase the availability of calcium, soluble iron, and zinc by large folds [1].

Then, after fermentation, the effect on amino acid and protein levels is controversial; for example, the concentration of available methionine, tryptophan, and lysine increases in cornmeal [1]. Likewise, fermentation of cereals such as millet, maize, sorghum, and other grains significantly enhances the quality of protein in addition to the levels of lysine [103]. However, while investigating sorghum kisra bread's nutritional content, no increase has been observed in lysine values, but there is an increase in methionine and tyrosine [104]. The tryptophan content increased while the manufacturing of uji, whereas it was also measured that there was a significant drop in lysine content [8]. It shows that the fermentation effect on foods' nutritional content is changeable, while the evidence for enhancement is significant. Fermentation in food results in improvement or enhances the flavor, texture, taste, aroma, and, most importantly, the product's shelf life.

Different volatile compounds are produced during cereals' fermentation, which generates a mixture of complex flavors in the products [102]. In addition to flavors, aroma-producing compounds like butyric acid and diacetyl acetic acid also upgrade the appeal of fermented cereal food products [8].

The most important ingredient used in traditional fermented foods and beverages prepared in most world regions is cereals (such as wheat, rice, wheat, sorghum, or corn). Some of these are used as significant foods in the human diet, whereas others are utilized as spices, colorants, breakfast, and beverages. In maximum cereal-based fermented products, fermentation is done with natural or mixed cultures such as bacteria, fungi, and yeasts. During fermentation, these microorganisms act parallel or in a sequential manner, changing dominant microbiota [1].

Species of *Bacillus*, *Lactobacillus*, *Micrococcus*, *Streptococcus*, *Pediococcus*, and Leuconostoc are the common fermenting bacteria. Moreover, *Cladosporium*, *Aspergillus*, *Trichothecium*, *Paecilomyces*, *Fusarium*, and *Penicillium* are the common fungal genera found in fermented products [3]. The predominant yeast species is *Saccharomyces*, used in the fermentation of alcohols [105]. These factors are responsible for developing fermenting microorganisms in fermented foods/beverages pH, water activity, concentration of salt, temperature, and food matrix composition. The most common microbes mediate food fermentation through L.A.B. [2,106]. This type of lactic acid fermentation provides enhanced nutritional value, safety, a longer shelf life of the fermented food products, and wide acceptability [107]. During natural fermentation in cereals, when cereals are cleaned and soaked in water for a few days, a succession of natural microbiota takes place, which is dominated by a large number of L.A.B.s.

In this fermentation type, amylases produce sugars fermented by lactic acid bacteria as a source of energy. Apart from fermentation, other steps like reducing size, salting, or heating also contribute to the final products [107].

Aguirre and Collins described L.A.B. as a broad group of catalase-negative, Gram-positive, non-motile, and non-sporing cocci and rods. They utilize fermentative carbohydrates to form lactic acid. Based on hexose sugars' utilization, the pathways are divided into two groups, i.e., homofermentative and heterofermentative [108].

In homofermentative pathways, the sole end or primary product of glucose fermentation by some lactic acid bacteria such as *Streptococcus*, *Pediococcus*, *Lactococcus*, and some lactobacilli is lactic acid. In contrast, in heterofermentative pathways, microorganisms such as *Leuconostoc*, *Weisella*, and some lactobacilli, the end products are ethanol, lactate, and $CO_2$ [107]. Lactic acid fermentation technology has also been confirmed for its preservative role in some cereals. L.A.B.-mediated antibiosis was characterized by the formation of hydrogen peroxide, antibiotics, and organic acids.

The production of organic acids by L.A.B. creates stressful conditions for spoilage-causing microorganisms present in cereals by reducing the pH to below 4.0 [63]. The acid action on the bacterial cytoplasmic membrane results in an antimicrobial effect, as it hinders the maintenance of the membrane potential, affecting active transport. Besides organic acid production, L.A.B. also produces hydrogen peroxide with flavin nucleotides, which rapidly react with oxygen and oxidize reduced nicotinamide adenine dinucleotide (NADH). It can also accumulate catalase enzyme (as true catalase is not present in the L.A.B. to degrade hydrogen peroxide) and inhibit several microorganisms [63]. In high tannin-containing cereals, lactic acid fermentation can also reduce tannin levels in some cereal crops to enhance iron absorption [18]. Fermentation through L.A.B. also contributes to viricidal activity [109] and antitumor properties [110].

Most of the legume products produced or consumed in Asia and Africa were fortified with cereals for improving the overall quality of proteins in the fermented product because legumes can provide a rich amount of lysine but are deficient in sulfur-containing amino acids [1]. However, cereals are rich in methionine and cysteine but are deficient in lysine [111].

### 4.8. Presence of Biogenic Amines in Juices and Vegetables Fermented with Lactic Acid Bacteria

Biogenic amines (B.A.) contain amines with low molecular mass and good biological efficacy and are found mostly in beverages and food items [112]. Over-intake of B.A. leads to several vasoactive symptoms and/or psychoactive effects [113].

Food spoilage during controlled or spontaneous fermentation usually leads to an increase in the concentration of B.A., where factors like temperature, pH, oxygen, or sodium chloride content adversely affect the formation of B.A. [114].

In many European countries, shredded white cabbage, popularly known as sauerkraut for its sensorial properties and nutritional value, is preserved by lactic acid fermentation [115]. Sauerkraut is produced in three steps, which are characterized by microorganisms that produce B.A. [115]. Some of these include *Leu. mesenteroides*, *Lactobacillus* sp., and *P. cerevisiae*.

In a study conducted by Kalac et al., in 121 samples of sauerkraut, the mean concentrations of B.A. present were found to be 50, 146, and 174 mg/kg for cadaverine, putrescine, and tyramine, respectively [116]. Histamine values were below 2 mg/kg in 44% samples, while in 19%, it was above 10 mg/kg. The study by Karovicova, and Kohajdova found that bottled and pasteurized sauerkraut juices (Germany) contained a large amount of putrescine up to 694 mg/dm$^3$ [117]. In another study done by Kalac et al., (2000a) the amounts of tyramine, putrescine, cadaverine, and spermidine were determined in different laboratory prepared sauerkrauts after 6 months of storage [118]. A similar study for B.A.'s determination was also done in lactic acid fermented cabbage juices of Germany [119]. In the carrot and red beet produced by lactic acid fermentation, the concentrations of cadaverine, histamine, putrescine, spermidine, and tyramine were found in the range of 1–15 mg/kg [120].

Kalac et al., prepared low B.A. content sauerkraut with certain conditions like a temperature of 15–20 °C for initial fermentation and pasteurization as soon as acidity and pH reach 9–10 g/kg and 3.8–4.0, respectively, to preserve them from bacteria [116]. Kolesarova (1995) [121] also found that the amount increases with pH values ranging from 3.6–3.8, thus suggesting pasteurization is stopped when the pH goes below 4, as at this stage yeast activity such as *P. cerevisiae* starts, increasing the amount of histamine to 200 mg/kg. Another study done by Kalac et al. found that sauerkrauts inoculated with a single *L. plantarum* or mixed culture of *L. casei*, *L. plantarum*, *E. faecium Pediococcus* sp. lead to the production of low concentrations of tyramine, putrescine, and cadaverine [118].

All the factors decreasing B.A. levels in sauerkraut production contribute to the initial stoppage of contamination by monogenic bacteria from shredded cabbage, the production of machines for shredding, silos, and transporters. However, Kalac et al., also reported the resistance of amine-negative inoculants to contamination [118].

## 5. Nutritional Value of Fermented Dairy Products

The values of fermented dairy products viz. sour milk, yogurt, dahi, kumiss, acidophilus milk, and other similar milk products are much more in demand due to their enhanced nutritional content compared to simple milk. Proteins, vitamins, carbohydrates, and some fat quality and quantity change in fermented milk, whereas the composition of minerals remains the same [122] (Figure 3). The quality of sour milk is determined by the fermenting microorganisms and the substances formed during milk souring's biochemical reactions. The substances include alcohol, lactic acid, antibiotics, carbon dioxide, and vitamins [123].

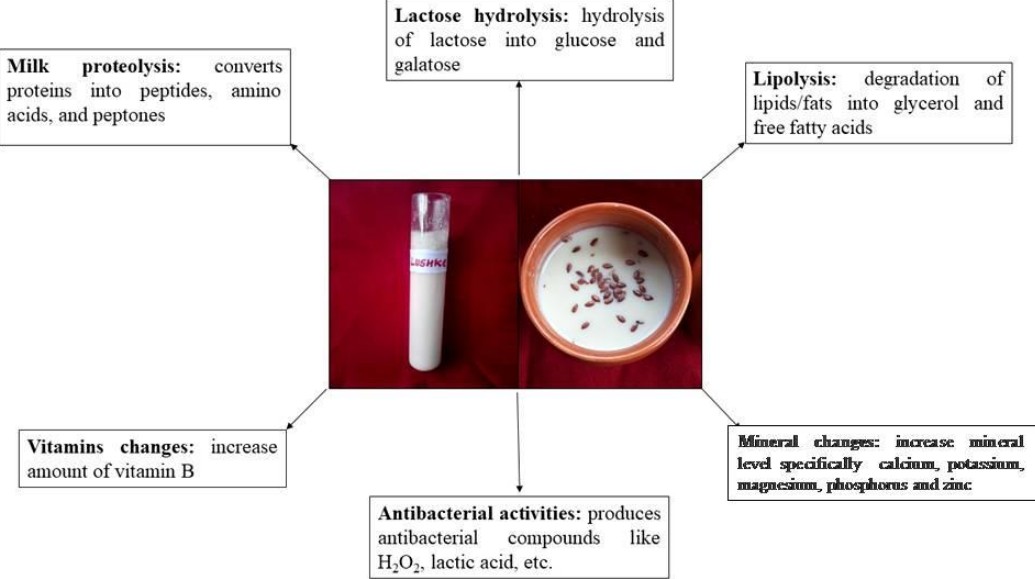

**Figure 3.** Changes in fermented dairy products during fermentation.

The biochemical processes mentioned below are beneficial for increasing the nutritional value of fermented milk:

(a)　Milk proteolysis

Proteolysis in milk causes the breakdown of proteins into peptides, amino acids, and peptones, which offer an increased amount of amino acids like methionine, leucine, phenylalanine isoleucine, tyrosine, tryptophan, valine, and threonine, which are essential A.A.s providing particular advantages, especially to physically weak persons. Proteolysis takes place with the help of Exo and endo peptides of L.A.B. [124]. The amount of protein increases from 85.4 to 90% in fermented milk such as dahi, yogurt, kefir, etc., and has higher digestibility for proteins due to the precipitation of milk by lactic

acid into fine curd, leading to increased nutritional value. The amounts of free amino acids, especially proline, phenylalanine, lysine, isoleucine, cysteine, and arginine, increase during fermentation and storage of fermented milk products. Due to proteolysis in milk and biochemical changes in milk protein, milk products become very dietetic [125].

(b)    Lactose hydrolysis

Lactose hydrolysis in milk is carried out by some bacteria present in the milk. The hydrolysis of lactose produces 0.6–0.8% glucose, 16–20% galactose, and 45–50% lactose compared to an average 5% lactose in milk. L.A.B. cause the hydrolysis of lactose through the production of β-galactosidase. Hydrolysis of lactose is vital for lactic acid production, which lowers the bowel's pH, inhibiting the putrefaction microorganisms from growing. Also, lactic acid is necessary for the absorption of calcium and organoleptic properties [126].

(c)    Lipolysis

The process of homogenization decreases the size of the fat globules, making them digestible [127]. Lipolysis results in physiological effects due to the increase in free fatty acid content by lactic acid bacteria.

(d)    Vitamins changes

The vitamin content in fermented milk depends on the bacterial culture present in it. Most vitamin B groups, especially riboflavin, thiamin, and nicotinamide, are increased two-fold, whereas vitamins B1, B2, and ascorbic acid decrease approximately by half via utilization by the growing bacteria in milk [128].

(e)    Antibacterial activity

The bactericidal action in fermented milk depends on the growing bacteria's antibiotic activity, such as lactobacilli in yogurt and other compounds producing antibacterial properties like hydrogen peroxide, lactic acid, bacteriocins, and antibiotics [129].

(f)    Mineral changes

There is increased and higher bioavailability of minerals in fermented milk, especially in calcium, potassium, zinc, magnesium, potassium iodide, and phosphorus caused by lactic acid bacteria during and after fermentation due to the process of fermentation and acidity [130].

## 6. Biochemical Changes in Meat-Based Fermented Food Products

*Lactobacillus*, *Streptococcus*, *Pediococcus*, *Leuconostoc*, *Lactococcus*, and *Enterococcus* are some of the facultative/obligate anaerobes belonging to gram-positive and acidogenic (lactic acid) bacteria that are used to metabolize saccharides with different degrees of efficiency into alcohols, lactic acid, lipids, amino acids, and aliphatic compounds (Figure 4) [131].

These organisms perform three functions simultaneously in fermented sausages, i.e., producing nitric oxide by reducing nitrate and nitrite, responsible for the cured color when combined with myglobin, and reducing pH by producing DL-lactic acid from glucose through anaerobic glycolysis [131].

Sausage incubation at optimum temperature with facultative anaerobic conditions causes rapid growth of the L.A.B. converting simple sugars into lactic acid and reducing the pH. A postmortem range of 4.5–7 μmol/g is not sufficient to lower down the pH; thus, simple sugars are added as a substrate for L.A.B., bringing pH to 4.6–5. Reference [132] used 0.62 g glucose/kg of meat to reduce the pH by 0.1 [132].

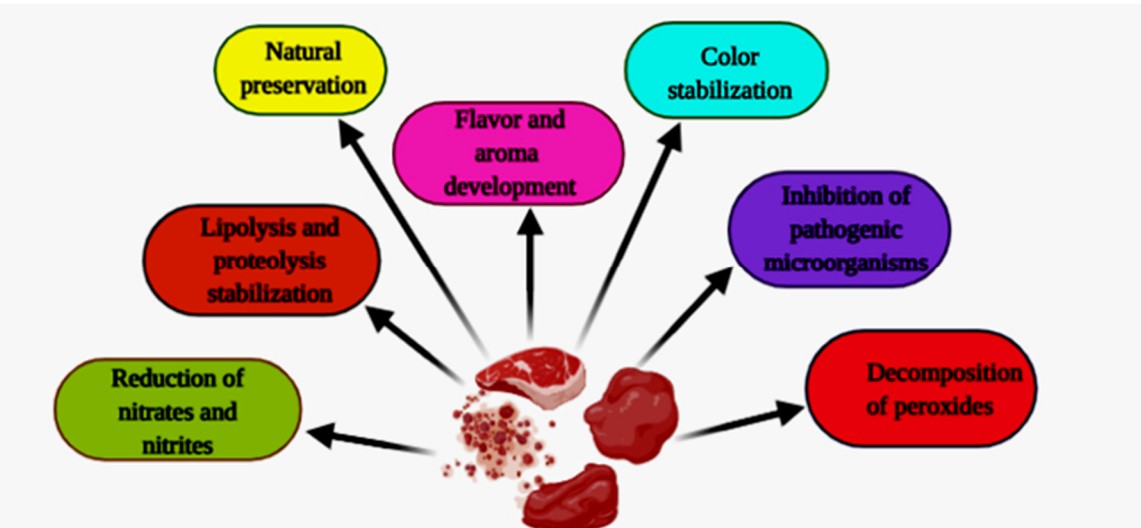

**Figure 4.** Quality changes in fermented meat products.

The primary mechanism for lactic acid metabolism includes fermentation of carbohydrates coupled with required degrees of phosphorylation.

The pathway of homofermentative lactic acid fermentation is through the Embden-Meyerhof-Parnas Pathway with lactic acid as an end product that leads to lactic acid as the sole end product, producing a sharp, tangy taste. Heterofermentative lactic acid bacteria also produce lactic acid through the phosphoketolase/6-phosphogluconate pathway, but it also liberates a small amount of acetic acid (approximately 10%), which is one of the causes of an off taste from hexoses [133].

The essential factors in lactic acid fermentation are time, temperature, and relative humidity; for example, higher water activity ($a_w$) and temperature aids in faster growth of L.A.B. and reducing the pH level. Smoke can contribute thousands of aroma and antimicrobial compounds such as organic acids (e.g., acetic, formic, butyric, isobutyric, and propionic acids), carbonyls, and phenols (antioxidants), contributing to the coagulation of surface proteins and inhibition of microorganisms [134].

The pH change is determined by the conversion of ammonia and lactate into lactic acid. This formation is done by adding carbohydrates produced from glycerol fermentation by bacteria and ammonia generated from amino acid fermentation. Acetic acid is also formed in this process of fermentation. Also, oxygen utilized through the metabolism process and other conditions influence the type of microorganisms and their metabolism, which affects the number of fermented carbohydrates and the production of lactose [135].

Cathepsin D (muscle's endogenous enzyme) breaks down myofibrillar proteins into polypeptides with low pH. Protein proteolysis produces volatile and non-volatile flavors in fermented sausages, resulting in free amino acids and peptides. Based on the literature cited by Fadda et al., the breakdown of flavor- and aroma-producing compounds like myofibrillar and sarcoplasmic proteins is done with the help of *L. plantarum* and *L. casei*. For the development of the structure of fermented sausages, these myofibrillar proteins are more critical than sarcoplasmic proteins [136].

In fermented meat products, lipolysis is caused by microbial enzymes of muscle tissues, i.e., both endo- and exoenzymes [137], and the oxidation of fatty acids results in the formation of alkanes, aldehydes, ketones, and alcohols [138].

Hydrolytic and oxidative changes, usually related to unsaturated fats and are autocatalytic, are responsible for the liberation of aromas and flavors (both good and bad flavors). There is a slight increase in thiobarbituric acid reactive substances (T.B.A.R.S.) in non-vacuum conditions, whereas it remains stationary in vacuum packages. Lipolytic enzymes naturally present in meat are responsible for releasing fatty acids, and in semi-dry fully cooked products, the activity is minimal. Smoking and mold ripening can be used to reduce oxygen by direct consumption and reduction in light penetration.

The sausage diameter also affects biochemical reactions involving greater diameter sausages (anaerobic) due to less oxidation. The concentration of glucose does not influence the maximum lipase activity; however, it is essential for rapid lipase production [139].

To date, 200 volatile compounds have been identified from fermented meat items that produce aroma. These compounds are also produced from smoking, spices, diacetyl (0.1 ppm), acetoin, acetaldehyde (0.7 to 1.5 ppm), and carbohydrate metabolism. For example, vinegar aroma is produced from acetic acid, while acetoin and diacetyl produce a butter-like aroma within the range of 1300–1400 ng/g. Phenylalanine breaks down into aldehydes, and methyl aldehyde produces fruity and malty aroma, whereas methyl acids contribute to a sweaty and cheesy aroma. Degradation of lipids produces alkanes, alkenes, and straight-chain aldehydes, green in color, metallic, fruity, and rancid in flavor. Ethanol (55 ppm) and 3 methyl-butanal yield a malty flavor and fruity mushroom/malty aroma, while methyl ketones are linked with *Staphylococcus carnosus* and result in a musty, fruity, and cheese-like aroma. Also, hexanal and methyl ketones produced from linoleic acid oxidation result in a green color and a metallic, fruity, rancid, musty, and cheese-like aroma. 2-nonanone produced from ketones by 2-pentanone yields green color, fruity, and mushroom-like aromas like fruits and mushrooms. Ketones produced from bacteria and chemical fermentation are encouraged by mold growth, mostly ethyl esters metabolized from ethanol. Fruity flavors in fermented sausages are contributed by esters [140].

Hydro peroxides, ketones, aldehydes, and other end products are produced by lipid oxidation, chemical or enzymatic. Factors influencing oxidation include oxygen content, the amount of unsaturated fatty acids connected to glycerol, prooxidants (e.g., metals, salt, etc.), and antioxidants (e.g., spices, nitrite, etc.). Oxidation is also affected by a rancid aroma, and different microorganisms that influence the oxidative pathways encountered [141].

The primary distinguishing factor between fermented and non-fermented sausages is the nonvolatile short-chain fatty acids. Reference [142] reported different ranges of organic acids like lactic acid, gluconic acid, acetic acid, and pyruvic acid [142]. Free fatty acids liberated from different lipase levels also correlate with the flavor complex, such as acid taste non-volatiles are correlated with D-lactate and acetate. The pH and aroma-producing acidic compounds are affected by non-protein nitrogen. Kasankala et al. reported that ATP metabolites (e.g., I.M.P., hypoxanthine), proteolytic activity due to raw materials containing cathepsin D-like enzymes, the proteolytic activity of ammonia and amines, sausages with long drying phase, acids (such as leucine, isoleucine, valine), aromatic branched aldehydes, and alcohols also influence taste [143].

Branched-chain amino acids in sausages have catabolites such as branched-chain alcohols, aldehydes, and methyl acids. In a study done by Demeyer et al., triglycerides with the help of both bacterial and endogenous enzymes liberate long-chain free fatty acids in the range of 27–37 mg/g, where the raw material and length of the ripening period determines the rate and amount of liberated free fatty acids and have little flavor [144].

## 7. Conclusions

Around the globe, for thousands of years, fermented foods have been part of the human diet due to the modifications in their natural form contributing to enhanced flavor and high-profile nutritional properties, without much information and knowledge of microbial functionality. With an extensive review of potentially cognitively influential changes in fermented foods, this paper may promote microbial and biochemical changes in fermented foods in a comprehensive manner that helps to understand the overall beneficial effect of microorganisms on fermented foods.

**Author Contributions:** R.S.: Collection of the information, planning, and drafting of the manuscript; P.G.: Data curation and formatting of the manuscript; S.K.: Conceptualization, analysis, review editing, and acquisition of funding. P.K. and S.K.B.: Analysis, review, and editing. All authors approved the final submitted version of the manuscript.

**Funding:** This research was funded by National Mission of Himalayan Studies (N.M.H.S.), grant number GBPNI/NMHS-2018-19/HSF27-05/156 and APC was funded by Shoolini University.

**Acknowledgments:** The authors would like to acknowledge the National Mission of Himalayan Studies (N.M.H.S.), Ministry of Environment, Forest and Climate Change (MoEFCC), New Delhi, and Shoolini University for providing a suitable environment for this research study. The authors would also like to thank the support of the Scientific Writing Cell of Shoolini University for language editing and formatting of the manuscript.

**Conflicts of Interest:** There is no conflict of interest in the present manuscript.

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
