# Peer review of "Microbial Fermentation and Its Role in Quality Improvement of Fermented Foods"

_fermentation, doi:10.3390/fermentation6040106_

Round 1

Reviewer 1 Report

A review concerning fermented foods with focus in the role of the microorganisms to the improvement of the original materials is of interest. However the main goal is much compromised because not only a good Review paper is not just a collect of data to be presented in one same paper (do not present the data as a list, tell a story about that data) but mainly due to the English that must be revised and improved. Some sentences are incorrect. For example, in lines 65-67, you do not mean to say that yeasts are mainly responsible for the isolation of the yeasts but that those genera are the main representing the yeasts isolated.

Lines 75-76: “This enhancement of nutrition in fermented foods are due to fermenting microorganisms by three different ways and they are as follows (Table 1):” I did not understand this sentence and the relation with the data presented in Table 1.

Line 62: “enzymes that assist indigestion.” You mean digestion!

Fig. 1 is not presented in the text and it is not discussed.

The word microflora, used in lines 68, 282, 293, is not used anymore, it is more correct to use microbiota.

Table 1: The name of the species is Weissella confusa and not confuse.

Table 1: Check and correct some species names that incorrectly begins with a capital letter.

Lines 162-168: Use l or L for liter as used in line 166 (ml). If L is choose then ml should be changed to mL.

Lines 169-178 and 179-187: The text in these lines are just a list of products where antioxidant activity or peptides where found. An integrative approach should be presented to these two topics.

Line 203 and 222 – Writing Bacillus subtilis natto or Bacillus natto is taxonomically incorrect. There is no species or subspecies named natto. Natto is the name of a strain.

Lines 302-304: lactobacilli it is not a genus name and should not be presented in the italic format and with a capital word. Present it as in line 397.

Lines 283-287: Apply the missing italic format to the genera names.

Lines 334, 335, 345, 348, 415, 416: Add a space before the unit in the cases that is missed.

References:

-        In several cases the italic format is missing in the name of the genus and species, for microorganisms and plants.

-        Some journals names are in the format presenting the complete name and others in the format with abbreviated names.

-        In vitro an in vivo also should be presented in the italic format

-        In references 28 and 34 the name of the journal is totally in capital letters.

Keywords – instead of presenting “fermenting organisms” and “Microorganism”, since only microorganisms (bacteria, yeasts, and filamentous fungi) are mentioned, change to “fermenting microorganisms”

Author Response

Response to reviewer’s comments:

Reviewer

Comments/Suggestions

Authors reply

Reviewer 1

A review concerning fermented foods with focus in the role of the microorganisms to the improvement of the original materials is of interest. However, the main goal is much compromised because not only a good Review paper is not just a collect of data to be presented in one same paper (do not present the data as a list, tell a story about that data) but mainly due to the English that must be revised and improved. Some sentences are incorrect. For example, in lines 65-67, you do not mean to say that yeasts are mainly responsible for the isolation of the yeasts but that those genera are the main representing the yeasts isolated.

The incorrect sentences has been modified and corrected.

Lines 75-76: “This enhancement of nutrition in fermented foods are due to fermenting microorganisms by three different ways and they are as follows (Table 1):” I did not understand this sentence and the relation with the data presented in Table 1.

The sentence has been corrected and the table has been included at the appropriate place

Line 62: “enzymes that assist indigestion.” You mean digestion!

“enzymes that assist indigestion” has now been changed to “enzymes that assist in digestion” in the revised manuscript

Fig. 1 is not presented in the text and it is not discussed.

Fig 1 has been annotated and briefly described

The word microflora, used in lines 68, 282, 293, is not used anymore, it is more correct to use microbiota.

The word microflora has now been changed to microbiota in revised manuscript

Table 1: The name of the species is Weissella confusa and not confuse.

The name of the species Weissella confuse  has now been corrected in the revised manuscript

Table 1: Check and correct some species names that incorrectly begins with a capital letter.

The corrections has been made in the revised manuscript

Lines 162-168: Use l or L for liter as used in line 166 (ml). If L is choose then ml should be changed to mL.

The corrections has been made in the revised manuscript

Lines 169-178 and 179-187: The text in these lines are just a list of products where antioxidant activity or peptides where found. An integrative approach should be presented to these two topics.

The detailed integrative approach have been added in the mentioned topics

Line 203 and 222 – Writing Bacillus subtilis natto or Bacillus natto is taxonomically incorrect. There is no species or subspecies named natto. Natto is the name of a strain.

The corrections has been made in the revised manuscript

Lines 302-304: lactobacilli it is not a genus name and should not be presented in the italic format and with a capital word. Present it as in line 397.

The corrections has been made in the revised manuscript

Lines 283-287: Apply the missing italic format to the genera names.

The missing italic format to the genera names have been done

Lines 334, 335, 345, 348, 415, 416: Add a space before the unit in the cases that is missed.

The corrections has been made in the revised manuscript

References:

In several cases the italic format is missing in the name of the genus and species, for microorganisms and plants.

The corrections has been made in the revised manuscript

Some journals names are in the format presenting the complete name and others in the format with abbreviated names.

The corrections has been made in the revised manuscript

In vitro and in vivo also should be presented in the italic format

The corrections has been made in the revised manuscript

In references 28 and 34 the name of the journal is totally in capital letters.

The corrections has been made in the revised manuscript

Keywords – instead of presenting “fermenting organisms” and “Microorganism”, since only microorganisms (bacteria, yeasts, and filamentous fungi) are mentioned, change to “fermenting microorganisms”

The corrections has been made in the revised manuscript

Reviewer 2 Report

The work entitled “Microbial fermentation and its role in quality improvement of fermented foods”, authors are Ranjana Sharma, Prakrati Garg, Pradeep Kumar, Shashi Kant Bhatia, and Saurabh Kulshrestha, is highly interesting and covers almost all fields in the problem of improvement of fermented food quality. The key microbial players in food fermentation are indicated with explanation of their impacts on fermented product properties, and crucial biochemical compounds generated at microbial fermentation are discussed. The work is comprehensive, well-structured, and confirmed with 137 references.  

Major revisions

The aim of this review is not clear. The topics discussed are not novel. Fermenting microorganisms are known, fermenting enzymes are described, and products of microbial metabolism at fermentation are known. What novel and original information is presented in the work? Maybe authors can stress on some novel approaches or novel knowledge obtained during last 5-10 years, and compare their review with other close reviews to show novelty and originality.

Minor revisions

p1, line 5, Title – there is no author after “and 2,*

p1, line 23 – “and antimicrobial” (space)

p1, line 44 – opinion that cheese has better taste than milk seems to be subjective and very personal (many people like cheese but do not like milk)

p1–2 – It is recommended to numerate paragraphs devoted to advantages of fermented food. It is more correct from a grammar point of view and makes text more readable.

p2, line 64 – “microorganisms” (together)

p2, line 65 – “Kluyveromyces” (one letter is lost)

p4, Table 1, and throughout the entire text – Authors should check first appearance of microorganisms in the text. At first appearance of a Latin name (full name including genus name and species epithet) in the text (starting from Introduction and until the References) it must be written in full, for example Lactococcus lactis, further it must be shortened as L. lactis at all other appearances until the end of the text.

p9, line 245 – “source for carbohydrates” or “source-for-carbohydrates” (spaces)

p10, lines 283–286 – Latin names must be italicized

p12, capture for Figure 3 – during what?

Summary. Overall recommendation: Reconsider after major revision.

Author Response

Response to reviewer’s comments:

Reviewer

Comments/Suggestions

Authors reply

Reviewer 2

The work entitled “Microbial fermentation and its role in quality improvement of fermented foods”, authors are Ranjana Sharma, Prakrati Garg, Pradeep Kumar, Shashi Kant Bhatia, and Saurabh Kulshrestha, is highly interesting and covers almost all fields in the problem of improvement of fermented food quality. The key microbial players in food fermentation are indicated with explanation of their impacts on fermented product properties, and crucial biochemical compounds generated at microbial fermentation are discussed. The work is comprehensive, well-structured, and confirmed with 137 references.  

Revised Manuscript submitted with all the suggestions incorporated

Major revisions:

The aim of this review is not clear. The topics discussed are not novel. Fermenting microorganisms are known, fermenting enzymes are described, and products of microbial metabolism at fermentation are known. What novel and original information is presented in the work? Maybe authors can stress on some novel approaches or novel knowledge obtained during last 5-10 years, and compare their review with other close reviews to show novelty and originality.

This review article focuses on the nutritional enhancement of fermented foods by microorganisms. So according to the theme of the manuscript, the novelty in terms of some major components has been described in the biological components for example:  in case of antioxidants activity, peptide production, etc.

Minor revisions:

p1, line 5, Title – there is no author after “and 2,*

The corrections has been made in the revised manuscript

p1, line 23 – “and antimicrobial” (space)

The corrections has been made in the revised manuscript

p1, line 44 – opinion that cheese has better taste than milk seems to be subjective and very personal (many people like cheese but do not like milk)

The sentence has been modified for better understanding

p1–2 – It is recommended to numerate paragraphs devoted to advantages of fermented food. It is more correct from a grammar point of view and makes text more readable.

The paragraphs have been numerated for making it more convenient to read

p2, line 64 – “microorganisms” (together)

The corrections has been made in the revised manuscript

p2, line 65 – “Kluyveromyces” (one letter is lost)

The corrections has been made in the revised manuscript

p4, Table 1, and throughout the entire text – Authors should check first appearance of microorganisms in the text. At first appearance of a Latin name (full name including genus name and species epithet) in the text (starting from Introduction and until the References) it must be written in full, for example Lactococcus lactis, further it must be shortened as L. lactis at all other appearances until the end of the text.

The appearance of microorganisms in the text and tables have been checked and corrected

p9, line 245 – “source for carbohydrates” or “source-for-carbohydrates” (spaces)

The corrections has been made in the revised manuscript

p10, lines 283–286 – Latin names must be italicized

The latin names have been italicized

p12, capture for Figure 3 – during what?

The caption has been corrected

Summary. Overall recommendation: Reconsider after major revision.

The final draft has been reconsidered carefully

Reviewer 3 Report

Manuscript ID: fermentation-972493
Title: Microbial fermentation and its role in quality improvement of fermented foods
Authors: Ranjana Sharma, Prakrati Garg, Pradeep Kumar, Shashi Kant Bhatia, Saurabh Kulshrestha
Probiotic Delivery through Non-Bovine Milk and Milk Products: Trends,
Novelties and Benefits https://www.mdpi.com/journal/fermentation/special_issues/probiotic_benefits

The review article does not clearly summarize previous published studies showing the current state of understanding on the topic of microbial fermentation to produce foodstuff nor provides a systematic analysis of the literature on the topic. General speaking the review article needs to be deeply revised and focused on the theme of microbial fermentation. I suggest authors to present what a technologist wants with the fermentation brought about by epiphytic microorganisms or by selected microorganisms inoculated with a specific purpose, depending on the raw material to ferment. Also the different types of fermentation, microorganisms generally used, substrates utilized (sugar, acids, peptides…depending on the raw material inoculated), the metabolites produced in each type of raw material, etc, must be presented.  The article is not accurately structured: sometimes it addresses aspects related to microbial activity, in others it addresses aspects of the raw material (cereals, dairy, meat….). The conclusions are quite insubstantial.  

Other comments/suggestions

Line 64: Change “micro organisms” to microorganisms

Line 65: I have some doubts about fitting the Aerococcus within the LAB group. 1st because this genus does not have the status GRAS since some of the species may be the causative agent in some infections. The genus Aerococcus was formed to harbor a group of Gram‐positive, microaerophilic, catalase‐negative coccus‐shaped bacteria, isolated from air and dust. The use of 16S rRNA gene sequencing discriminate them from closely related streptococci and enterococci. For more details I recommend the reading of the following article: https://onlinelibrary.wiley.com/doi/10.1002/9781118655252.ch7

Line 66: Please rephrase the statement “ Debaryomyces, Kluyeromyces, and Saccharomyces are mainly responsible for the isolation of the yeasts”

Lines 66-67: please clarify the meaning of that statement “ Molds involving the Geotrichium, Mucor, Penicillium, and Rhizopus species [8, 9]”

Lines 69-71: Figure 1 needs to be verified and improved; The genus Lactobacillus encompasses homo- and heterofermentative species. Therefore, it is necessary to clarify the group of Lactobacillus that use this metabolic path;

Lines 71-72: the figure capitation it is not correct: probably sugar metabolism by yeasts and LAB

Lines 74- 79: Please rewrite and clarify the concepts of the following sentences: i) “It has been known that fermented foods are more nutritious than their unfermented counterparts [13]; ii) This enhancement of nutrition in fermented foods are due to fermenting microorganisms by three different ways and they are as follows (Table 1); iii) Microorganisms being catabolic also breaks down more complex compounds. Moreover, they are anabolic and are also responsible for the synthesis of complex vitamins as well as other growth factors [14]”.

Lines 102-103: There are several aspects regarding this table that needs to be modified. According to the legend of Table 1 Authors present the “Microorganisms involved in preparation of fermented foods”. The content of the table it is not in agreement with the sentence in lines 75-76;

Lines 283- 286:  Species must be in italic. Please check all the text;

Lines: 367 – Something is missing in Figure 3 legend. “Changes in fermented dairy products during”. The figure also needs revision. As an example: mineral changes: no mineral changes

Author Response

Response to reviewer’s comments:

Reviewer

Comments/Suggestions

Authors reply

Reviewer 3

The review article does not clearly summarize previous published studies showing the current state of understanding on the topic of microbial fermentation to produce foodstuff nor provides a systematic analysis of the literature on the topic. General speaking the review article needs to be deeply revised and focused on the theme of microbial fermentation. I suggest authors to present what a technologist wants with the fermentation brought about by epiphytic microorganisms or by selected microorganisms inoculated with a specific purpose, depending on the raw material to ferment. Also the different types of fermentation, microorganisms generally used, substrates utilized (sugar, acids, peptides…depending on the raw material inoculated), the metabolites produced in each type of raw material, etc, must be presented. The conclusions are quite insubstantial. The article is not accurately structured: sometimes it addresses aspects related to microbial activity, in others it addresses aspects of the raw material (cereals, dairy, meat….). 

The review article summarizes the importance of microorganisms in the area of fermented foods. How microorganisms are responsible for the enhancement of various biological components, is discussed in detail in this manuscript whereas fermented foods according to the raw materials/substrates used were also discussed focussing the microbes and enzymes involved in them. Additional data in case of important components i.e. antioxidants and peptides was added and the revised manuscript has been restructured accurately.

Line 64: Change “micro organisms” to microorganisms

The corrections has been made in the revised manuscript

Line 65: I have some doubts about fitting the Aerococcus within the LAB group. 1st because this genus does not have the status GRAS since some of the species may be the causative agent in some infections. The genus Aerococcus was formed to harbor a group of Gram-positive, microaerophilic, catalase-negative coccus-shaped bacteria, isolated from air and dust. The use of 16S rRNA gene sequencing discriminate them from closely related streptococci and enterococci. For more details I recommend the reading of the following article: https://onlinelibrary.wiley.com/doi/10.1002/9781118655252.ch7

In many of the scientific papers Aerococcus has been placed in the LAB group. However it has also been observed that this bacteria is opportunistic in nature which is capable of causing infection. So, due to the risk involved, it has been removed from the LAB group in the manuscript

Line 66: Please rephrase the statement “ Debaryomyces, Kluyeromyces, and Saccharomyces are mainly responsible for the isolation of the yeasts”

The corrections has been made in the revised manuscript

Lines 66-67: please clarify the meaning of that statement “Molds involving the GeotrichiumMucor, Penicillium, and Rhizopus species [8, 9]”

The corrections has been made in the revised manuscript

Lines 69-71: Figure 1 needs to be verified and improved; The genus Lactobacillus encompasses homo- and heterofermentative species. Therefore, it is necessary to clarify the group of Lactobacillus that use this metabolic path;

The figure and the caption has been corrected

Lines 71-72: the figure caption is not correct: probably sugar metabolism by yeasts and LAB

The figure caption has been corrected accordingly

Lines 74- 79: Please rewrite and clarify the concepts of the following sentences: i) “It has been known that fermented foods are more nutritious than their unfermented counterparts [13]; ii) This enhancement of nutrition in fermented foods are due to fermenting microorganisms by three different ways and they are as follows (Table 1); iii) Microorganisms being catabolic also breaks down more complex compounds. Moreover, they are anabolic and are also responsible for the synthesis of complex vitamins as well as other growth factors [14]”.

The sentences have been corrected carefully

Lines 102-103: There are several aspects regarding this table that needs to be modified. According to the legend of Table 1 Authors present the “Microorganisms involved in preparation of fermented foods”. The content of the table it is not in agreement with the sentence in lines 75-76

The legend of Table 1 and the content in the lines have been modified accordingly

Lines 283- 286:  Species must be in italic. Please check all the text;

The text has been checked thoroughly

Something is missing in Figure 3 legend. “Changes in fermented dairy products during”. The figure also needs revision. As an example: mineral changes: no mineral changes

The figure legend has been corrected and the changes in mineral content was also done

Round 2

Reviewer 1 Report

  1. Line 131 – The word process is repeated “… the process of fermentation process …”
  2. Lines 149 and 152 - Line 149 – “ … been identified, L.A.B. is …”; “… Line 152 - involve lactic acid bacteria (L.A.B.) …” Consider that L.A.B., standing for lactic acid bacteria, should be presented as in line 152 but the first time in line 149 (the first time it is mentioned). Then in line 152 use L.A.B. it is enough.
  3. Lines 231, 715 – Since the meaning of L.A.B. was presented previously there is no need to do it again in these sentences; use lactic acid bacteria or L.A.B.
  4. 1 – Lactobacillus and Saccharomyces should be in the italic format
  5. Lines 165 and 166 – “was mentioned” “was also discussed”. I think that the use of present tense is more adequate.
  6. Line 169 - Correct “assist indigestion” to assist in digestion
  7. Line 175 – Figure 1.- The legend presented is “Sugar metabolism by L.A.B. and yeast”. However it is more correct to say Sugar metabolism by Lactobacillus and Saccharomyces as representatives of L.A.B. and yeasts.
  8. Table 1. – In Deb. Tamari correct the capital T to t; in L. Plantarum correct the capital P to p
  9. A reference should not be used as a noun, it is not correct to start or finish a sentence with a number in brackets: Line 443 - [49] determined antioxidant; Line 449 - assessed by [50].; Line 722 - [109] described; 761 - conducted by [117] in; 765 - [118] the study found; 779 - done by [119] found; 785 - However, [119] also; 881 - Cited by [137]; 924 - [144] reported; 929 - done by [145]
  10. Lines 780, 781, 882 - in L. Plantarum correct the capital P to p
  11. Line 478 – Apply the italic format to Bacillus
  12. Lines 707 and 718 - The word flora (as microflora) is not used anymore, it is more correct to use microbiota.
  13. Lines 708-711: Apply the missing italic format to the genera names.
  14. Line 731 – “LAB-mediated” since L.A.B. as been used in the remaining text it should also be applied here.
  15. Line 842 – Correct … that arssssse used to
  16. Table 1 – The genera and species names should be carefully revised. For example, A. is used for Aspergillus (A. oryzae) and Acetobacter (A. pasteurianus) and that it is taxonomically incorrect. Another example is Acetobacter xylinum that it is misspelling and should be Acetobacter xylinus (check the information in https://lpsn.dsmz.de/species/acetobacter-xylinum): Acetobacter xylinum (sic) (Brown 1886) Yamada 1984 homotypic synonym, misspelling; Acetobacter xylinus corrig. (Brown 1886) Yamada 1984 homotypic synonym, validly published.

Author Response

Response to the comments of Reviewer 1:

Reviewer

Comments/Suggestions

Authors reply

Reviewer 1

Line 131 – The word process is repeated “… the process of fermentation process …”

The sentence have been corrected

Lines 149 and 152 - Line 149 – “ … been identified, L.A.B. is …”; “… Line 152 - involve lactic acid bacteria (L.A.B.) …” Consider that L.A.B., standing for lactic acid bacteria, should be presented as in line 152 but the first time in line 149 (the first time it is mentioned). Then in line 152 use L.A.B. it is enough.

The corrections has been made in the revised manuscript

Lines 231, 715 – Since the meaning of L.A.B. was presented previously there is no need to do it again in these sentences; use lactic acid bacteria or L.A.B.

The corrections has been made in the revised manuscript

1 – Lactobacillus and Saccharomyces should be in the italic format

The corrections has been made in the revised manuscript

Lines 165 and 166 – “was mentioned” “was also discussed”. I think that the use of present tense is more adequate.

The corrections has been made in the revised manuscript

Line 169 - Correct “assist indigestion” to assist in digestion

The corrections has been made in the revised manuscript

Line 175 – Figure 1.- The legend presented is “Sugar metabolism by L.A.B. and yeast”. However it is more correct to say Sugar metabolism by Lactobacillus and Saccharomyces as representatives of L.A.B. and yeasts.

The corrections has been made in the revised manuscript

Table 1. – In Deb. Tamari correct the capital T to t; in L. Plantarum correct the capital P to p

The corrections has been made in the revised manuscript

A reference should not be used as a noun, it is not correct to start or finish a sentence with a number in brackets: Line 443 - [49] determined antioxidant; Line 449 - assessed by [50].; Line 722 - [109] described; 761 - conducted by [117] in; 765 - [118] the study found; 779 - done by [119] found; 785 - However, [119] also; 881 - Cited by [137]; 924 - [144] reported; 929 - done by [145]

The corrections has been made in the revised manuscript

Lines 780, 781, 882 - in L. Plantarum correct the capital P to p

The corrections has been made in the revised manuscript

Line 478 – Apply the italic format to Bacillus

The corrections has been made in the revised manuscript

Lines 707 and 718 - The word flora (as microflora) is not used anymore, it is more correct to use microbiota.

The corrections has been made in the revised manuscript

Lines 708-711: Apply the missing italic format to the genera names.

The corrections has been made in the revised manuscript

Line 731 – “LAB-mediated” since L.A.B. as been used in the remaining text it should also be applied here.

The corrections has been made in the revised manuscript

Line 842 – Correct … that arssssse used to

The corrections has been made in the revised manuscript

Table 1 – The genera and species names should be carefully revised. For example, A. is used for Aspergillus (A. oryzae) and Acetobacter (A. pasteurianus) and that it is taxonomically incorrect. Another example is Acetobacter xylinum that it is misspelling and should be Acetobacter xylinus (check the information in https://lpsn.dsmz.de/species/acetobacter-xylinum): Acetobacter xylinum (sic) (Brown 1886) Yamada 1984 homotypic synonym, misspelling; Acetobacter xylinus corrig. (Brown 1886) Yamada 1984 homotypic synonym, validly published.

The corrections has been made in the revised manuscript

Reviewer 2 Report

All reviewer's comments were taken by authors into consideration. After revesion, the idea has become clearer, and it is understandable that the focus is done on role of microorganisms in food quality and safety improvement.

There are two small remarks concerning Latin names:

  • please, check and correct: species epithet must be written in lowercase letters/ For example, L. plantarum is correct, and L. Plantarum is wrong;
  • lines 708-721 - please, check this paragraph, all Latin names must be in italics.

Author Response

Response to the comments of Reviewer 2:

Reviewer

Comments/Suggestions

Authors reply

Reviewer 2

 All reviewer's comments were taken by authors into consideration. After revesion, the idea has become clearer, and it is understandable that the focus is done on role of microorganisms in food quality and safety improvement.

There are two small remarks concerning Latin names:

·         please, check and correct: species epithet must be written in lowercase letters/ For example, L. plantarum is correct, and L. Plantarum is wrong;

The corrections has been made in the revised manuscript

Lines 708-721 - please, check this paragraph, all Latin names must be in italics.

The corrections has been made in the revised manuscript

Reviewer 3 Report

According to author's reply most of the recommendations have been accepted thus I trust after all this work this review  article can be accepted for publication in the journal.

Author Response

Response to the comments of Reviewer 3:

Reviewer

Comments/Suggestions

Authors reply

Reviewer 3

According to author's reply most of the recommendations have been accepted thus I trust after all this work this review article can be accepted for publication in the journal.

I would like to thank you for your valuable suggestions to make this manuscript better and taking this into consideration.
